# Chronic Pain-Associated Cardiovascular Disease: The Role of Sympathetic Nerve Activity

**DOI:** 10.3390/ijms24065378

**Published:** 2023-03-11

**Authors:** Christian A. Reynolds, Zeljka Minic

**Affiliations:** 1Department of Emergency Medicine, Wayne State University School of Medicine, 540 E Canfield St., Detroit, MI 48201, USA; 2Department of Biotechnology, University of Rijeka, 51000 Rijeka, Croatia

**Keywords:** sympathetic nerve activity, chronic pain, cardiovascular disease

## Abstract

Chronic pain affects many people world-wide, and this number is continuously increasing. There is a clear link between chronic pain and the development of cardiovascular disease through activation of the sympathetic nervous system. The purpose of this review is to provide evidence from the literature that highlights the direct relationship between sympathetic nervous system dysfunction and chronic pain. We hypothesize that maladaptive changes within a common neural network regulating the sympathetic nervous system and pain perception contribute to sympathetic overactivation and cardiovascular disease in the setting of chronic pain. We review clinical evidence and highlight the basic neurocircuitry linking the sympathetic and nociceptive networks and the overlap between the neural networks controlling the two.

## 1. Introduction

Chronic pain syndromes affect 20–30% of the world’s population [1] and there is a clear link between chronic pain and the development of cardiovascular disease, which exists across a spectrum of chronic pain syndromes, including back or pelvic pain, neuropathic pain, and fibromyalgia. A meta-analysis that included 25 large observational studies of patients across the spectrum of chronic pain syndromes, found a significant association between chronic pain and cardiovascular disease [2]. A second meta-analysis found a similar relationship between chronic musculoskeletal pain and cardiovascular disease [3]. Multiple longitudinal studies have also reported an association between chronic pain and the development of cardiovascular disease [4,5,6,7,8,9,10,11,12,13,14,15,16], which (for many of the studies) remained significant after adjustment for cardiovascular risk factors [4,9,11,12,13,14,15,16]. The largest of these studies recently reported on the relationship between chronic pain and cardiovascular disease among 475,171 participants in the UK Biobank [16]. The results of this large study indicate that participants with chronic localized pain and chronic widespread pain had a significantly increased risk for future incidence of myocardial infarction, heart failure, stroke, cardiovascular mortality, and composite cardiovascular disease. The study also provides direct evidence of a dose–response relationship between chronic pain severity and cardiovascular morbidity. As summarized in Table 1, multiple studies have identified increased risk among chronic pain sufferers for myocardial infarction, angina, arrhythmia, coronary artery disease, hypertension, stroke, heart failure, and cardiovascular mortality.

Scientific evidence linking pain to the cardiovascular system dates to the early 1900s when Sir Charles Sherrington observed that experimental pain triggers acute cardiovascular responses [34,35]. Over the decades, a wealth of literature reporting on pharmacological, neuroanatomical, electrophysiological, and behavioral data indicated that a highly conserved neural network regulates both the sympathetic nervous system (SNS) and pain perception. We hypothesize that chronic pain-related changes within these neural networks contribute to increased cardiovascular disease in chronic pain sufferers. In addition to reviewing available clinical evidence, we emphasize the basic neurocircuitry driving cardiovascular disease in chronic pain syndromes and highlight the causative role of the SNS in precipitating chronic pain-associated cardiovascular comorbidities. While SNS overactivation in chronic pain patients may be partially explained by the direct effects of nociceptive stimulation on sympathetic preganglionic neurons [36,37,38], we highlight the central neural network regulating sympathetic nerve activity (SNA) and pain perception, which we propose contributes to SNS overactivation in the setting of chronic pain.

## 2. Sympathetic Nervous System and Cardiovascular Disease

The SNS is critical for general cardiovascular homeostasis [39] as it exerts direct actions on heart rate and cardiac contractility as well as venous capacitance, arteriolar resistance, and blood volume (via sodium and water balance in the kidneys). However, excessive SNS activation increases cardiovascular morbidity and mortality [39,40]. Acute overactivation of the SNS can manifest as adverse cardiac events including: ventricular arrhythmias, myocardial infarction, atrial fibrillation, stroke, and Takotsubo cardiomyopathy [40]. Chronic overactivation of the SNS leads to hypertension, ischemic heart disease, heart failure, and renal failure [39]. 

In chronic pain, cardiovascular indices that indicate SNS overactivation, including increased blood pressure and heart rate, have been well documented [41,42,43,44,45]. Additionally, SNS overactivation contributes to the process of atherosclerosis by inducing platelet activation [46,47,48] and promoting mechanical injury to the vascular endothelial cells because of increased blood pressure and flow velocity. At the level of the heart, the ensuing atherosclerosis manifests with coronary artery disease and can trigger myocardial infarction. Furthermore, chronic, and excessive sympathetic drive to the heart (i) limits myocardial oxygen delivery via coronary vasoconstriction, while also (ii) increasing myocardial oxygen demand because of increased energy utilization. Ultimately, this combination of reduced oxygen delivery and increased oxygen demand leads to myocardial ischemia, which can result in angina, arrhythmia, or even heart failure [49]. 

## 3. Basic Neurocircuitry of the Sympathetic Nervous System 

Sympathetic preganglionic neurons are cholinergic neurons located within the intermediolateral (IML) cell column of the spinal cord. These neurons innervate pre- and para-vertebral ganglia where they synapse with adrenergic postganglionic neurons. One exception is the adrenal gland, which is directly innervated by sympathetic preganglionic neurons and drives the systemic release of epinephrine. The regulation of sympathetic preganglionic neuron-firing is influenced by various interconnected neuronal networks, including: (i) the central autonomic network providing descending projections from the brain, and (ii) the intraspinal network consisting of propriospinal neurons and spinal interneurons. The level of sympathetic activity targeting the effector organs is ultimately determined by the balance of excitatory and inhibitory inputs to sympathetic preganglionic neurons in the IML. 

The central autonomic network serves an essential physiological role in the coordinated, real-time adaptations of SNS in response to external and internal stimuli. Brain regions, which comprise the central autonomic network include: (i) brainstem centers (both medulla and midbrain), (ii) diencephalon, and (iii) cortical sites; all of which work in a coordinated manner to influence sympathetic tone and cardiovascular function [50]. Arguably, to date, most is known about the role of brainstem centers in controlling SNA. It was over 100 years ago when the rostral ventrolateral medulla (RVLM) was first identified as a primary regulator of sympathetic tone [51,52] as bilateral lesioning or pharmacological blockade of the ventrolateral medulla produced decreases in sympathetic activity and arterial hypotension [51,52,53]. Since then, many studies have substantiated this initial observation and highlighted the RVLM as one of the most important sources of descending excitatory drive to sympathetic preganglionic neurons located in the IML [54,55,56,57,58]. Additionally, the RVLM and adjacent brainstem centers are involved in baroreflex processing, which is critical for quick reflex-mediated changes in SNA (Figure 1). Afferent neurons innervating the carotid sinus and aortic arch carry afferent, baroreceptor-related information, to the medulla, via the vagal and glossopharyngeal nerves. During baroreceptor loading, as occurs during increases in blood pressure, the nucleus of the solitary tract (NTS) receives excitatory glutamatergic input from these afferent neurons and activates inhibitory GABAergic neurons within the caudal ventrolateral medulla (CVLM). The CVLM, in turn, inhibits RVLM neurons and decreases the excitatory drive to the sympathetic preganglionic neurons located in the IML. It is important to note that in addition to decreasing excitatory RVLM drive, baroreceptor loading increases descending inhibitory drive to sympathetic preganglionic neurons located in the IML [59,60,61]. Such descending inhibitory projections may arise from nuclei of the CVLM, the reticular formation, or the locus coeruleus (LC), which also receive direct projections from barosensitive NTS neurons. While the central neural networks involved in driving descending inhibition of IML neurons are not well defined, in response to baroreceptor loading, SNA can be reduced to levels similar to those observed following ganglionic blockade (Figure 2, panel A), which is well below the levels recorded following bilateral pharmacological inhibition of RVLM [62] or spinal cord transection (Figure 2, panel B) [63,64,65,66]. These data suggest that: (i) intraspinal networks alone can maintain baseline SNA tone and blood pressure and that (ii) the inhibition of SNA in response to baroreceptor loading (phenylephrine infusion) originates from within the supraspinal centers, possibly the LC or other brainstem centers.

In addition to reflex regulation, cognitive processing and emotion regulation are important drivers of sympathetic activity involving the central autonomic network. Classical cardiovascular responses to emotional stress (fight and flight response or defense reaction) are produced by disinhibition of the paraventricular and dorsomedial nuclei of the hypothalamus and drive sympathoexcitatory responses and increases in blood pressure (Figure 3) [67,68,69]. Similarly, increases in SNA are observed during strenuous cognitive tasks, i.e., mental stress and they seem to be gender specific [70,71]. To sustain increases in sympathetic activity in response to emotional stress, the medullary baroreflex circuitry is reset or overridden by the higher brain structures, which helps to prepare the body for action [72,73,74,75]. Similarly, baroreflex resetting occurs during perceived mental stress and highlights the important role of perception in controlling SNS [76]. Such baroreflex resetting is believed to involve direct cortico-medullary projections converging in the NTS to attenuate the medullary baroreflex processing [77,78]. Emotional and mental stress-induced increases in sympathetic activity involve various cortical brain regions some of which have been implicated in tonic control of sympathetic activity at rest [79,80,81,82]. For example, electrical stimulation of discrete areas of the insular cortex can markedly increase blood pressure by approximately 50 mmHg and increase heart rate by about 40 bpm [83]. In addition to the insular cortex, other important cortical regions are the anterior cingulate cortex and prefrontal cortex. These regions form an integrative network where the insular and anterior cingulate cortex send projections to the hypothalamus and the amygdala [80]. The amygdala as a central feedback regulator sends inhibitory projections to the NTS, thereby deactivating the medullary baroreflex mechanism and activating the hypothalamic nuclei, which drive sympathoexcitatory responses associated with stress reaction [84].

In addition to the descending excitatory and inhibitory projections controlling sympathetic activity outlined above, sympathetic preganglionic neurons within the IML receive extensive input from intraspinal networks [85,86]. Spinal circuits can contribute both to baseline and reflex responses in sympathetic activity. Resting sympathetic activity and blood pressure are maintained (albeit at a lower level), following cervical or thoracic spinal cord transection and they can be further reduced by ganglionic blockade [63,64]. Although tonically low, sympathetic activity can be markedly increased following the activation of spinal sympathetic pathways. The spinal sympathetic reflex circuitry is similar in composition to the spinal motor reflex circuits that drive muscle contraction in response to muscle stretch or cutaneous stimulation, and like spinal motor reflexes, the spinal sympathetic reflexes are exaggerated by spinal cord transection [87,88,89,90,91]. The spinal motoneurons are controlled by the descending corticothalamic projections, which exhibit a tonic inhibitory drive to the motoneurons [92,93]. Similarly, we speculate that descending inhibitory projections arising from or above the level of the brainstem, modulate sympathetic drive. Tonic sympathoinhibitory projections are important regulators of overall sympathetic activity and they can arise from various central structures including, the CVLM [94,95], raphe nuclei [96], and the spinal cord [97]. These structures have been shown to inhibit sympathetic activity tonically [95,98,99]. This inhibition likely involves both direct bulbospinal projections [100,101,102], and projections to the RVLM [94,95,98]. Additional baroreceptor-independent inhibitory projections descend via the dorsolateral funiculus and reduce spinal sympathetic reflex activation [103,104]. Finally, the inhibitory influences on the sympathetic preganglionic neurons can be exerted by the spinal interneurons located within and around the IML. To that end, electrical stimulation of the central autonomic area of the spinal cord triggered inhibitory postsynaptic potentials within the IML, and these potentials were blocked by the GABA receptor blockade, suggesting that spinal GABA-ergic interneurons modulate sympathetic preganglionic transmission and influence tonic sympathetic activity [105].

## 4. Basic Neurocircuitry of Nociception

In this section, we try to briefly summarize functional neural circuitry involved in the detection, transmission, and central integration of nociceptive signals. Based on the presence of thermal, chemical, or mechanical stimuli, the signal is detected by specialized receptors (nociceptors) expressed on peripheral terminals of small thinly myelinated or unmyelinated A-delta and C fibers of primary afferent neurons, respectively [106,107]. These afferents carry nociceptive information into laminae I-III of the dorsal horn of the spinal cord where they undergo complex processing. The excitatory and inhibitory interneuronal circuitry integrates incoming information and transmits it to projecting neurons for relay to the brain. Nociceptive information travels to the brain via the contralateral spinothalamic tract synapsing within the ventromedial and mediodorsal nuclei of the thalamus [108]. The nociceptive information also reaches the medulla and brainstem centers via the spinoreticular and spinomesencephalic tracts and the hypothalamus via the spinohypothalamic tract [109]. For the processing of nociceptive signals, especially important is the NTS, which integrates ascending noxious and non-noxious information and provides an interface between sensory and autonomic outflows. The NTS has dense projections with other sub-cortical brain regions such as periaqueductal gray, nucleus raphe magnus, and the hypothalamic nuclei, all of which are also involved in autonomic processing [108]. Finally, nociceptive information is subjected to processing at the cortical level where pain is conceptualized through localization and intensity discrimination. Regarding its localization, somatosensory cortices I and II are mostly responsible for determining the position of nociceptive stimulus while anterior cingulate cortex is involved in assigning an affective component to the pain [110,111]. Furthermore, the insular cortex can encode the intensity, and localization of the painful stimulus as it relates to the generation of moods and feelings [112]. It is the strong relationship between the anterior cingulate cortex and the insula that forms a link between pain, autonomic, and motor systems through the formation of motivation and emotion related to pain. 

Interestingly, numerous monosynaptic descending projections have been described in the literature that simultaneously project to second-order neurons in the dorsal horn and to sympathetic preganglionic neurons in the IML. Table 2 highlights brain regions with monosynaptic descending projections to both the dorsal horn and the IML, and the effect of these projections on nociception and SNA. Inhibitory and excitatory projections can arise from all three integrative levels: (i) the brainstem, (ii) the hypothalamus, and (iii) the cortex, with most regions having a common excitatory and/or inhibitory effect on SNA and pain. Importantly, the NTS and LC, elicit strong inhibition of both second-order sensory neurons and sympathetic preganglionic neurons, which reduce nociceptive processing and SNA, respectively.

## 5. The Relationship between Blood Pressure and Acute Pain Sensitivity

In pain-free individuals, there is a linear, inverse (negative) relationship between blood pressure and acute pain sensitivity [143,144,145,146,147,148,149,150,151,152,153]. Importantly, the diminished acute pain sensitivity association with elevated resting blood pressure involves the medullary baroreflex circuitry, reducing both SNA and pain processing in response to increases in systemic blood pressure. In animals, baroreceptor stimulation induces antinociception [154,155,156], and surgical denervation of baroreceptor afferents produces hyperalgesia [157,158,159]. In humans, stimulation of baroreceptors (e.g., via application of external suction to the carotid artery) reduces acute pain sensitivity [160,161,162,163]. Consistent with the notion that baroreceptor loading is greatest during systole and lowest during diastole, pain sensitivity to ultra-rapid electrical pain stimuli was found to be lowest during systole and the greatest during diastole [164,165]. Altogether, these studies indicate that in addition to reducing SNA, baroreceptor stimulation triggers descending pain inhibitory activity, which contributes to the inverse relationship between resting blood pressure and acute pain. 

In the setting of chronic pain, the relationship between blood pressure and pain sensitivity is dramatically altered. In contrast to the significant inverse (negative) relationship observed in pain-free individuals, chronic pain sufferers exhibit a direct (positive) relationship between resting blood pressure and acute pain [42,43,144]. Changes to the blood pressure-pain sensitivity relationship in the chronic pain setting directly contribute to SNS overactivation. A large retrospective study identified an increased prevalence of hypertension in chronic back pain patients and pain intensity was a significant predictor of hypertensive status [166]. This suggests that chronic pain may drive SNS overactivation and thereby promote cardiovascular morbidity and mortality. 

## 6. Chronic Pain as a Driver of SNS Overactivation

As introduced above, spinal sympathetic reflex arcs result in the direct activation of sympathetic preganglionic neurons within the IML by ascending sensory/nociceptive pathways. This likely represents the most straightforward neural mechanism by which chronic pain can drive SNS overactivation. Within the dorsal horn, sensory information is directed to: (i) interneurons, which terminate within the same spinal segment, (ii) propriospinal neurons, which terminate within a different spinal segment, and (iii) projecting neurons, which terminate within supraspinal structures. Activation of sympathetic preganglionic neurons occurs either via second-order interneurons, second-order propriospinal neurons, or second-order spinothalamic projecting neurons, which have extensive axon collaterals within the thoracic spinal cord [167]. Additional interneurons (third order) terminating within the intermediolateral cell column of the spinal cord are likely critical for sensory-sympathetic coupling, as second-order neurons (arising from the dorsal horn) rarely make direct connections with cell bodies of sympathetic preganglionic neurons [168]. Thus, in the setting of chronic pain, SNS overactivation may result directly from excess spinal nociceptive input via a spinal sympathetic reflex arc. 

A second mechanism by which chronic pain may increase sympathetic activity is secondary to chronic cognitive or emotional stress. Excessive sympathetic stress responses are associated with numerous human disorders, including post-traumatic stress disorder, cerebral palsy, traumatic brain injury, autism spectrum disorders, bipolar disorder, epilepsy, and Charcot–Marie–Tooth disease, among others [169,170,171,172], and can manifest with acute myocardial injury and/or increased susceptibility to sudden cardiac death [173,174,175]. Cardiovascular consequences of stress are similar in all mammals [67,176,177], and chronic or repetitive stress leading to increases in sympathetic activity is one proven cause of hypertension and heart failure [176,178]. Pain, especially when present in a chronic form, is a source of discomfort and substantial emotional suffering. In chronic orofacial pain patients, sympathetic responses to mental stressors are significantly greater than responses in pain-free, age and sex-matched controls [179]. Similarly, patients with chronic low back pain or chronic arthritic pain display a significantly higher baseline and stress-induced sympathetic (electrodermal) arousal than what is observed in pain-free controls [45,180,181]. 

## 7. Maladaptive Changes in the Neural Circuitry Leading to Sympathetic Overactivation and Chronic Pain

Our understanding of neuroplastic changes associated with chronic pain has been largely made possible by advancements in in vivo imaging using magnetic resonance imaging (MRI) techniques. Enhanced spatial resolution and contrast enable detailed structural morphology studies to be conducted in patients with chronic pain. Functional magnetic resonance imaging (fMRI) studies in humans have demonstrated the coordinated activation of several brain areas in response to noxious somatic and visceral stimuli, including the thalamus, anterior cingulate cortex, insular cortex, primary and secondary sensory cortices, prefrontal cortex, basal ganglia, cerebellum, and amygdala. This network of brain regions involved in both sensory discriminative and emotional-affective aspects of pain is termed the “pain matrix” [182]. Studies utilizing resting state fMRI revealed changes in network properties in chronic pain when compared to healthy controls. Specifically, atrophy within the dorsolateral prefrontal cortex (dlPFC) and increased activity of the medial prefrontal cortex (mPFC) have been observed in chronic back pain sufferers [183]. Similarly, a general shift from nociceptive to emotional circuits has been observed in chronic back pain [184], which may corroborate the involvement of emotional circuits in driving sympathetic overactivation in chronic pain. As mentioned above, these brain structures are also involved in the control of sympathetic activity both at baseline and during emotional stress. Thus, SNS overactivation among chronic pain sufferers likely involves an exaggerated excitatory drive to sympathetic preganglionic neurons as part of a chronic emotional stress response. 

Chronic pain is associated with plastic changes at every level of the neural axis including primary sensory endings, as well as spinal and supraspinal sites [185]. The changes within the supraspinal centers are particularly important as they may contribute to altered nociceptive processing and perception of pain. Given the anatomical and functional overlap between the central nociceptive and central autonomic networks, the aberrant changes that occur within the supraspinal centers may simultaneously manifest with dysregulated sympathetic control and chronic pain. Recent studies have found that chronic pain sufferers irrespective of the origin of pain, exhibit common brain signatures associated with the loss of gray matter in cortical and subcortical structures such as the prefrontal and insular cortex, orbitofrontal cortex, and pons [185]. Using connectivity analyses of resting state fMRI, chronic migraine sufferers exhibited reduced gray matter volume and reduced cortical thickness in brain regions involved in the affective processing of pain including dlPFC [186]. The cortical thickness was inversely associated with the intensity and duration of pain suggesting a potential causative relationship between structural changes and the clinical phenotype of chronic pain. Additionally, decreased connectivity of the prefrontal cortex and increased connectivity with the insular cortex were found to be correlated with the intensity of chronic pain [187]. Both the insular cortex and prefrontal cortex represent areas of the brain involved in the emotional processing that led to an increased drive to the central autonomic network. The baseline activity of dlPFC is elevated in chronic pain sufferers [188] and direct transcranial stimulation of dlPFC in healthy individuals was found to increase SNA and blood pressure [137]. Additionally, specific neuronal ensembles within the mPFC appear to be critical for processing nociceptive information and regulating pain chronicity [189]. These data suggest the reorganization of the cortical centers associated with chronic pain states may also promote sympathetic overactivation and the development of cardiovascular disease via increased descending excitatory drive to sympathetic preganglionic neurons in the spinal cord. 

Lastly, given that a common baroreceptor-sensitive, descending, inhibitory pathway regulates the spinal transmission of afferent nociceptive information and efferent sympathetic activity, it is conceivable that alterations within this central inhibitory network would simultaneously manifest with increased pain and increased sympathetic activity. Central noradrenergic pathways, particularly those mediated by alpha-2 adrenergic receptors (α2), are a crucial component of both the descending pain inhibitory system [190,191,192] and the descending pathway regulating SNS [60,193]. The contribution of α2 receptors to descending nociceptive inhibition is well documented [194,195,196,197,198,199] and the activation of central α2 receptors is important for the inhibition of sympathetic nerve activity in response to baroreceptor loading [59,61,193,200]. Multiple central structures, including the locus coeruleus (LC), are sources of descending pain-sympathetic modulation via α2 receptor activation [190,201,202,203,204], and likely contribute to descending inhibition of both pain and SNS. Therefore, alterations in this central neural network, which inhibits both spinal sympathetic preganglionic neurons and spinal transmission from afferent nociceptive neurons, would contribute to a concomitant increase in both pain and SNS activity. 

Within the NTS there are numerous glutamatergic and GABAergic terminals arising from interneurons or from cortical and hypothalamic nuclei, which facilitate or inhibit excitation of the NTS, (e.g., during baroreceptor loading). Notably, the amygdala, the bed nucleus of the stria terminalis, and the PVN all project to the NTS [205,206]. Projections from cortical and hypothalamic centers to the NTS are known to inhibit excitatory neurotransmission of barosensitive neurons and participate in the upward resetting of the baroreflex [207]. Thus, it is tempting to speculate that chronic maladaptive changes within the central neural network, resulting in enhanced inhibition of barosensitive neurons within the NTS, serve as another plausible mechanism by which both chronic pain and increased sympathetic activity would simultaneously manifest. 

## 8. Conclusions

There are many points along the neuraxis where nociceptive signals interact with SNS processing and alterations in these neural networks in the setting of chronic pain likely contribute to the SNS overactivation and development of cardiovascular disease. It remains unclear what neural processes contribute to SNS overactivation in the setting of chronic pain. Recent findings in human models have identified the structures within the prefrontal cortex that are involved in pain chronicity [189] and regulate sympathetic nerve activity and cardiovascular function [137]. These new findings indicate there are structural connections between chronic pain and cardiovascular function. Additionally, we highlight other direct and indirect neural networks that may play a role in driving chronic pain-associated cardiovascular disease. Future studies are needed to understand this relationship in more detail.

## Figures and Tables

**Figure 1 ijms-24-05378-f001:**
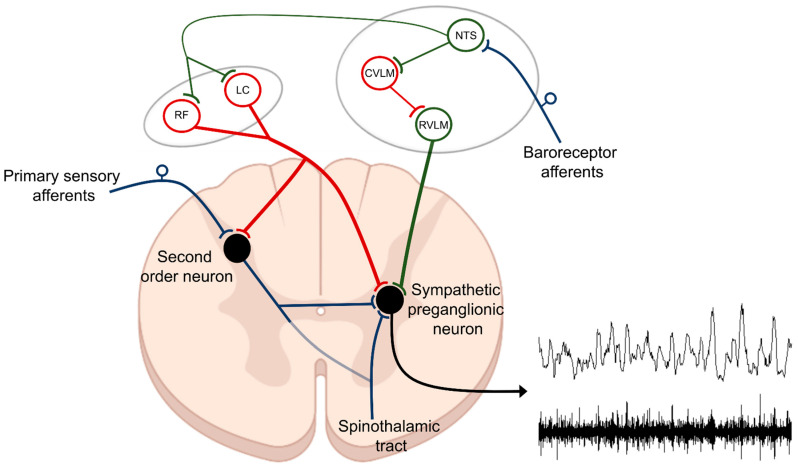
Control of sympathetic preganglionic neurons. Efferent sympathetic nerve activity (SNA) is determined by the net product of excitatory and inhibitory projections to sympathetic preganglionic neurons located within the intermediolateral (IML) cell column of the thoracic spinal cord. The rostral ventrolateral medulla (RVLM) is one of the main sources of descending excitatory drive (green). Additionally, second-order sensory neurons, relaying information from primary sensory afferents, are an important intraspinal source of excitatory drive to sympathetic preganglionic neurons. Descending inhibitory projections to sympathetic preganglionic neurons (red) arise from multiple brain regions including from the locus coeruleus (LC) and the reticular formation (RF). Much of the descending excitatory and inhibitory drive to sympathetic preganglionic neurons is regulated by neurons located within the nucleus of the solitary tract (NTS), which is the primary integrative site for baroreceptor and chemoreceptor afferent fibers which drive autonomic reflexes. Included above is an example of raw (lower tracing) and integrated (upper tracing) SNA recorded from postganglionic renal sympathetic nerve fibers in a rat as described in [63] (lower right).

**Figure 2 ijms-24-05378-f002:**
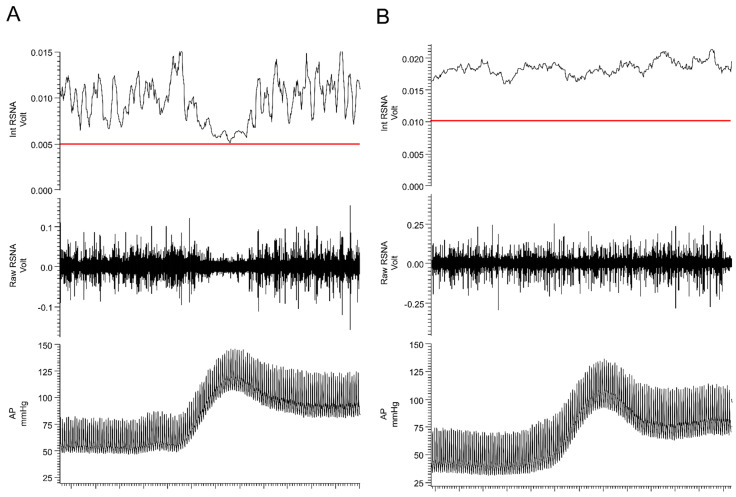
Baroreceptor loading strongly inhibits sympathetic nerve activity. Responses in arterial pressure (AP) and renal sympathetic nerve activity (RSNA) to intravenous administration of phenylephrine (40 µg/kg) in (**A**) spinal intact and (**B**) T4 spinal-transected rats. The red line in the top panel represents the level of recorded activity (zero) at the end of the experiment following ganglionic blockade (hexamethonium, 20 mg/kg). See text for additional information.

**Figure 3 ijms-24-05378-f003:**
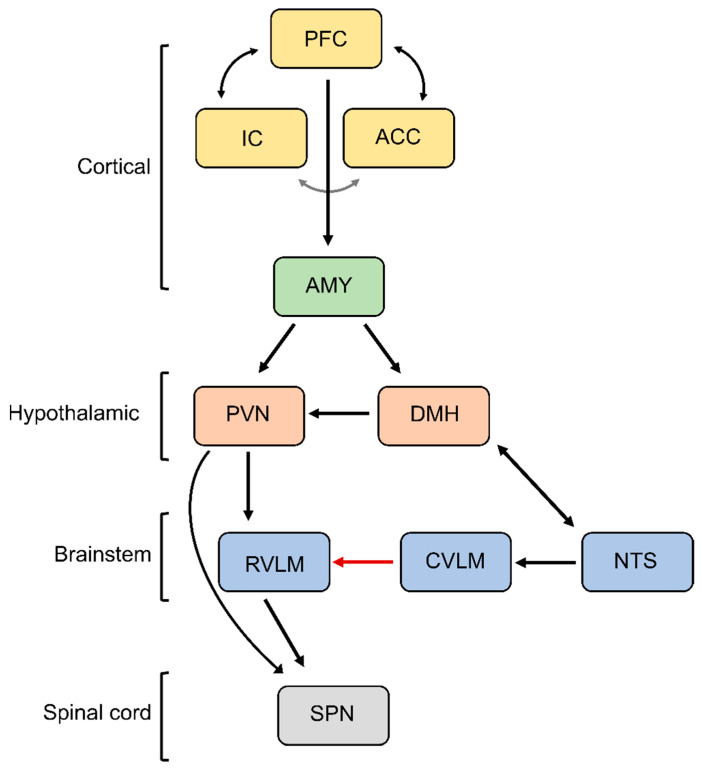
Central autonomic network. Depicted is the neural circuitry controlling SNA during emotional stress. PFC, prefrontal cortex; IC, insular cortex; ACC, anterior cingulate cortex; AMY, amygdala; DMH, dorsomedial hypothalamus; PVN, paraventricular nucleus; RVLM, rostral ventrolateral medulla; CVLM, caudal ventrolateral medulla; NTS, nucleus of the solitary tract; SPN, sympathetic preganglionic neurons. See text for additional information.

**Table 1 ijms-24-05378-t001:** Chronic pain-associated cardiovascular outcomes.

Cardiovascular Outcome	References
Myocardial Infarction	[10,14,16,17,18,19,20,21,22]
Angina	[14,17,18,22,23,24]
Arrhythmia	[20,25]
Coronary Artery Disease	[14,22,25,26,27]
Hypertension	[20,25,28,29,30]
Stroke	[10,16,17,19,20,21,29]
Heart failure	[16,20]
Cardiovascular mortality	[8,16,21,22,30,31,32,33]

**Table 2 ijms-24-05378-t002:** Overview of monosynaptic pathways descending to the spinal cord that influence both second-order sensory neurons and sympathetic preganglionic neurons.

Structure	Effect(s) on DH Neurons	Effect(s) on IML Neurons	References
**Hypothalamus**			
Paraventricular nucleus	Inhibition	Inhibition/Excitation	[113,114,115,116]
Arcuate nucleus	Inhibition/Facilitation	Inhibition/Excitation	[117,118,119]
Parabrachial nucleus	Inhibition	Excitation	[120,121,122]
**Brainstem**			
Nucleus of the solitary tract	Inhibition	Inhibition	[123,124,125,126]
Raphe magnus/pallidus	Inhibition/Facilitation	Inhibition/Excitation	[127,128,129,130]
Rostroventromedial medulla	Facilitation	Excitation	[131]
Locus coeruleus	Inhibition	Inhibition	[132,133,134]
Medullary reticular formation	Inhibition/Facilitation	Inhibition/Excitation	[135,136]
**Cerebral cortex**			
Frontal/parietal	Inhibition	Inhibition/Excitation	[137,138,139,140,141,142]

DH, dorsal horn; IML intermediolateral cell column.

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
