# Peer review of "Chronic Pain-Associated Cardiovascular Disease: The Role of Sympathetic Nerve Activity"

_ijms, 2023, doi:10.3390/ijms24065378_

Round 1

Reviewer 1 Report

In this manuscript, the authors are to highlight the role of over sympathetic nerve activity in chronic pain-associated cardiovascular disease. There is crosstalk between the neurocircuitry of nociception and sympathetic nerve system. The neurons that receive noxious information are involved in control of sympathetic activity. Chronic pain facilitates the excitatory drive to sympathetic preganglionic neurons and results in over activity of sympathetic nerve and cardiovascular disease. This is an interesting manuscript.

 1.    This manuscript is titled “Chronic pain-associated cardiovascular disease: the role of sympathetic nerve activity”, however, the author didn’t discuss what kind of cardiovascular disease (exception hypertension) is associated with chronic pain.

2.    The authors hypothesized that maladaptive changes within a common neural network regulating sympathetic nervous system and pain contribute to sympathetic overactivation and cardiovascular disease in the setting of chronic pain. What are the maladaptive changes within a common neural network?

3.    In pain-free individuals, there is a linear, inverse (negative) relationship between blood pressure and acute pain sensitivity. In humans, stimulation of baroreceptors (e.g. via application of external suction to the carotid artery) reduces acute pain sensitivity. Stimulation of baroreceptors will inhibit sympathetic nerve activity and decreases blood pressure. So, this experiment doesn’t display the inverse (negative) relationship between blood pressure and acute pain sensitivity. This manuscript is to discuss the role of sympathetic nerve activity, sometime, the blood pressure can’t represent sympathetic nerve activity. Blood pressure or sympathetic nerve may display a diverse array of heterogeneous reactions to nociception (PMID 27445972), not always an inverse (negative) relationship.

4.    Diagram will be useful for summarizing the relationship of chronic pain and sympathetic overactivation.

5.  Referencing recent articles assures the information up to date in this review manuscript. Such as Qi et al. found (2022, PMID: 36516746) neuroplastic alterations in dorsomedial prefrontal cortex involve in pain chronicity, while Sesa-Ashton (2022, PMID: 35559424) reported that dorsolateral prefrontal cortex regulates sympathetic nerve activity and cardiovascular function. These new findings indicate there are structural connections between chronic pain and cardiovascular function.

Reviewer 2 Report

although well-written this manuscripts ares

nothing significantly new to the field

Author Response

Thank you for your comments.

Reviewer 3 Report

No comments

Author Response

Thank you

Reviewer 4 Report

The review provides a comprehensive overview of the current understanding of the link between chronic pain and cardiovascular disease.The literature search appears to be comprehensive and includes a diverse range of relevant sources. The conclusion that sympathetic nerve activity plays a role in the association between chronic pain and cardiovascular disease is well-supported by the evidence presented in the review. The review is original in its contribution to the field by synthesizing current knowledge and highlighting areas for future research. The writing style is clear and concise, making the information accessible to a broad audience. However, I highly suggest the authors add several figures in the manuscript, which could help make the paper more readable and better understand the potential role of sympathetic nerve activity in chronic pain related cardiovascular disease.

minor concern: The font in this manuscript is not uniform

Reviewer 5 Report

This paper assumes the reader has an intimate knowledge of neuroanatomy and neurophysiology. Please include pictures so that a reader not deeply knowledgeable of nerve pathways can understand the paper better than its present condition.

Basic neurocircuitry - Brian is Brain ............... 3rd paragraph of same section - Similar is Similarly .......... which helps to prepare THE body....

Reviewer 6 Report

The review is devoted to chronic pain-associated cardiovascular disease caused by the activation of the sympathetic nervous system. 

The main criticism: there is a tiny section (number 2) in the review devoted to sympathetic nervous system and cardiovascular disease. The significant part of the manuscript describes basic neurocircuitry of the sympathetic nervous system and basic neurocircuitry of nociception which is a good idea but considering the title of the review more attention should be paid to the connection between chronic pain-associated cardiovascular disease and sympathetic nerve activity. It would be nice to have a list of cardiovascular diseases (and also a clear definition of what is considered a cardiovascular disease) associated with chronic pain with references provided (for example, there was no mention in the review about the connection of atherosclerosis with chronic pain). The authors should also clearly state possible the mechanism/s  connecting chronic pain, sympathetic nervous system and cardiovascular diseases. It was mentioned in the review in section 2 that sympathetic nervous system overactivation can cause increased blood pressure and heart rate, but I believe that all factors even potentially contributing to Cardiovascular Disease development due to proper or improper work of sympathetic nervous system caused by chronic pain should be listed.

The authors should mention in the conclusion their detailed view on the most beneficial for the field experiments which should be done to shed more light on the role of sympathetic nervous system as a connector between chronic pain and cardiovascular diseases.

The review lacks illustrations. I would suggest to create at least one figure listing pathological factors contributing to cardiovascular disease development caused by chronic pain and sympathetic nervous system.  

The list of abbreviations used would benefit the paper.

The format of reference number 1 and other references in the list of references used should be corrected according to MDPI guidelines.

The review can be accepted after major revision.

Round 2

Reviewer 6 Report

The manuscript has been improved now and can be accepted for a publication.